# Breaking Bad: Inflammasome Activation by Respiratory Viruses

**DOI:** 10.3390/biology12070943

**Published:** 2023-07-01

**Authors:** Julia A. Cerato, Emanuelle F. da Silva, Barbara N. Porto

**Affiliations:** 1Department of Medical Microbiology and Infectious Diseases, Rady Faculty of Health Sciences, University of Manitoba, Winnipeg, MB R3E 0J9, Canada; ceratoj@myumanitoba.ca (J.A.C.); dasilvae@myumanitoba.ca (E.F.d.S.); 2Biology of Breathing Group, Children’s Hospital Research Institute of Manitoba, Winnipeg, MB R3E 0J9, Canada

**Keywords:** respiratory virus, inflammasome, virus infection, antiviral response, disease pathogenesis

## Abstract

**Simple Summary:**

Inflammasomes are multiprotein complexes that sense endogenous damage stimuli and diverse microbial pathogens, including viruses. A growing body of evidence shows that inflammasome activation by respiratory viruses, such as influenza virus and coronaviruses, is a double-edged sword. While inflammasomes are important for viral clearance and tissue injury recovery, an uncontrolled inflammasome activation leads to hyperinflammation and severe disease. This review summarizes the up-to-date knowledge on how respiratory viruses activate inflammasomes and how it influences disease pathogenesis.

**Abstract:**

The nucleotide-binding domain leucine-rich repeat-containing receptor (NLR) family is a group of intracellular sensors activated in response to harmful stimuli, such as invading pathogens. Some NLR family members form large multiprotein complexes known as inflammasomes, acting as a platform for activating the caspase-1-induced canonical inflammatory pathway. The canonical inflammasome pathway triggers the secretion of the pro-inflammatory cytokines interleukin (IL)-1β and IL-18 by the rapid rupture of the plasma cell membrane, subsequently causing an inflammatory cell death program known as pyroptosis, thereby halting viral replication and removing infected cells. Recent studies have highlighted the importance of inflammasome activation in the response against respiratory viral infections, such as influenza and severe acute respiratory syndrome coronavirus 2 (SARS-CoV-2). While inflammasome activity can contribute to the resolution of respiratory virus infections, dysregulated inflammasome activity can also exacerbate immunopathology, leading to tissue damage and hyperinflammation. In this review, we summarize how different respiratory viruses trigger inflammasome pathways and what harmful effects the inflammasome exerts along with its antiviral immune response during viral infection in the lungs. By understanding the crosstalk between invading pathogens and inflammasome regulation, new therapeutic strategies can be exploited to improve the outcomes of respiratory viral infections.

## 1. Introduction

Respiratory viruses include an extensive range of viruses that infect the cells of the upper and lower respiratory tract. Most mild respiratory virus infections are restricted to the upper respiratory tract; however, children, the elderly, immunocompromised individuals, and those with chronic conditions are at high risk of developing a severe lower respiratory tract infection (LRTI). Adenovirus, rhinovirus, influenza virus, respiratory syncytial virus (RSV), parainfluenza virus, and coronaviruses are the most frequent respiratory viruses that cause severe LRTIs. Once in the lower respiratory tract, these viruses are recognized by epithelial cells and resident alveolar macrophages through pattern recognition receptors (PRRs) [1,2,3]. PRRs recognize viral genetic material and proteins and activate the immune system [4]. PRRs include Toll-like receptors (TLRs), DNA sensors such as cyclic GMP-AMP synthase (cGAS), and retinoic acid-inducible gene-I (RIG-I). Some PRRs, such as NOD-, LRR-, and pyrin domain-containing protein 1 (NLRP1), NOD-, LRR-, and pyrin domain-containing protein 3 (NLRP3), NLR family CARD domain-containing protein 4 (NLRC4), and absent in melanoma 2 (AIM2), recruit apoptosis-associated speck-like proteins (ASC), as well as caspase 1, to form the inflammasome [5]. The inflammasome is a multi-protein complex that consists of a sensor, an adaptor, and an effector molecule. The canonical activation of inflammasomes requires two signals: the priming signal (or signal 1) is provided by TLR activation or cytokine receptor signaling, which will induce the activation of NF-κB and the up-regulation of inactive pro-IL-1β and inflammasome components. The activation signal (or signal 2) is provided by various stimuli, including extracellular ATP, pore-forming toxins, and RNA viruses. These events lead to the oligomerization of the inflammasome complex through direct action or ASC interaction with inactive pro-caspase 1 [6,7]. After the formation of the inflammasome complex, caspase 1 cleaves pro-IL-1β/IL-18 to its mature form [8]. Caspase 1 can also cleave and activate gasdermin D, which induces pore formation in the plasma membrane and consequently IL-1β/IL-18 release and pyroptotic cell death [9].

Several respiratory viruses have been shown to activate each of the major inflammasomes (Table 1). For example, NLRP3 is activated in response to RSV, influenza virus, adenovirus, and coronavirus infection [10,11,12,13,14]. Some viruses are sensed by multiple inflammasomes, as is the case of influenza, adenoviruses, and rhinoviruses [15,16,17,18,19,20,21]. A growing body of evidence suggests that inflammasome activation by respiratory viruses is a double-edged sword, as it significantly contributes to both viral clearance and the induction of severe disease [14,18,20,22,23]. In this review, we summarize the recent progress in our understanding of how common and emerging respiratory viruses and their isolated proteins modulate the inflammasome. We highlight the beneficial antiviral effects of inflammasome activation by specific respiratory viruses and how inflammasome inhibition or hyperactivation can become detrimental to the host. Additionally, we discuss some recent experimental and clinical approaches to regulate its activity and potential unwarranted harmful effects.

## 2. Influenza Virus

Influenza viruses are still responsible for illness and deaths worldwide and still represent a threat to public health. Influenza A virus (IAV) is the most common viral activator of the NLRP3 inflammasome. It has been shown that influenza virus infection induces the NLRP3/caspase 1 pathway, with posterior secretion of IL-1β and IL-18 in bone marrow-derived macrophages [24]. In addition, the influenza virus activates the NLRP3 inflammasome in non-immune cells, such as lung fibroblasts and primary bronchial epithelial cells [17,25]. NLRP3 has been shown to sense viral RNA during IAV infection in mice [25]. IAV infection increased the expression of IL-1β in the bronchoalveolar lavage (BAL) fluid of wild type mice, but not NLRP3 deficient (Nlrp3^−/−^), caspase 1 deficient (Casp^−/−^), or ASC deficient (Asc^−/−^) mice, suggesting the importance of inflammasome proteins in the antiviral response [22,25]. In these studies, mice deficient in inflammasome proteins were more susceptible to IAV infection than wild type mice. The enhanced morbidity correlated with reduced inflammatory cytokines in BAL fluid, such as tumor necrosis factor (TNF) and IL-6. Interestingly, NLRP3 activation can be either protective or harmful depending on the stage of IAV infection. Blocking NLRP3 with MCC950 24 h post-infection resulted in accelerated weight loss and increased mortality in mice. On the other hand, delaying NLRP3 inhibition to later stages of IAV infection promoted a significant delay in mortality and the reduction of proinflammatory cytokines in the BAL of mice [26]. This study demonstrated that NLRP3 is crucial for virus clearance at early stages of infection but can induce exaggerated inflammation during later stages of infection.

Several viral proteins have been shown to activate innate sensors. Pore-forming viral proteins, known as viroporins, can alter ion concentrations in intracellular compartments and therefore initiate inflammasome activation. IAV viroporin M2 enables proton export across the trans-Golgi network, leading to alterations in ion flux and triggering NLRP3 inflammasome activation. M2 protein activity was required and sufficient to promote inflammasome activation by influenza in primed macrophages and dendritic cells [27]. However, influenza viruses have developed multiple strategies to evade recognition and antiviral responses by the host innate immune system. In particular, the IAV NS1 protein has been reported to bind to NLRP3 and inhibit assembly of the complex NLRP3-ASC-caspase 1 and hence the secretion of IL-1β [28]. Inflammasome-mediated cytokine secretion is crucial for the induction of virus-specific adaptive immunity and lung damage repair following IAV infection [29]. Therefore, NLRP3 inflammasome inhibition by the NS1 protein may be associated with IAV disease pathogenesis [29].

In addition to NLRP3, AIM2 inflammasome expression and activation have also been described for the influenza virus [20]. During infection, the AIM2 inflammasome is activated and plays a critical role in IAV-induced lung injury and mortality [20]. These data suggest that the AIM2 inflammasome may be a valuable therapeutic target for treating the severe inflammatory consequences of influenza virus infection.

Mutations in inflammasome components may result in aberrant activation and lead to macrophage activation syndrome (MAS) and cytophagocytosis. This induces excessive IL-1R signaling and drives autocrine inflammation. MAS and inflammasome-derived cytokines were determinant factors of lethal H1N1 and 1918 IAV infection [30,31]. Furthermore, sustained NLRP3 activation following IAV infection caused excessive inflammatory monocyte recruitment and lung pathology [32]. Surprisingly, NLRP3-deficient mice also presented lung damage and reduced IL-1β following IAV infection [25]. Together, these studies indicate that inflammasome activation must be precisely regulated to provide appropriate inflammatory responses and protection without causing damage to the host lung tissue.

## 3. Parainfluenza Virus

Human parainfluenza virus (HPIV) belongs to the paramyxovirus family. It is responsible for causing a spectrum of life-threatening respiratory diseases, such as bronchiolitis, pneumonia, and croup, especially during infancy and childhood [33,34]. The literature regarding inflammasome activation by HPIV is limited. HPIV activates TLR2 or TLR4 in macrophages, responsible for triggering antiviral immune responses and activating inflammasomes, mainly NLRP3 [4,35]. Human parainfluenza virus type 3 (HPIV3) has been shown to activate the NLRP3/ASC complex in macrophages, resulting in the production of mature and active IL-1β from HPIV3-infected cells [35]. TLR2 activation worked as the first signal, and the second signal was potassium efflux mediating inflammasome activation. Surprisingly, the same study found the HPIV3 C protein to be an antagonist of inflammasome activation, promoting proteasomal degradation of the NLRP3 protein and blocking inflammasome activation [35]. Additionally, HPIV2 blocks inflammasome function by inhibiting the activation of caspase 1 and the maturation of IL-1β in human monocytic THP-1 cells [36].

## 4. Human Metapneumovirus

Human metapneumovirus (HMPV) belongs to the *Pneumoviridae* family [37]. Several reinfections with this virus can occur throughout life, with the primary infections happening before 5 years of age. HMPV is one of the leading infectious agents responsible for respiratory diseases. Young children and the elderly population are at a greater risk of severe disease [37].

The literature regarding HMPV infection and the inflammasome is scarce. However, some evidence shows that NLRP3 inflammasome activation can be detrimental during the course of infection, being responsible for high levels of pro-inflammatory cytokines. An up-regulation of IL-1β and NLRP3 genes and over-expressed IL-18 levels were observed in the nasopharyngeal aspirates and plasma of children with a high severity score disease [38]. In addition to NLRP3-derived IL-1β and IL-18, HMPV infection also leads to the over-expression of TNF and IL-6 [38,39], which can worsen lung inflammation and disease outcomes following infection.

Le et al. (2019) correlated the HMPV small hydrophobic protein with NLRP3 inflammasome activation. The authors observed that mice infected with HMPV lacking this protein exhibited less inflammation, mortality, and lung histopathological scores, suggesting that this protein alone could be responsible for the morbidity and mortality caused by HMPV. Studying the role of NLRP3 in HMPV-infected mice, they discovered that inflammasome activation did not alter the viral load, and high levels of IL-1β led to more severe pathogenicity [38]. On the other hand, NLRP3 inflammasome-associated HMPV pathogenesis is IL-18 independent [39].

## 5. Respiratory Syncytial Virus

Respiratory syncytial virus (RSV) causes a huge disease burden in the infant, immunocompromised, and elderly populations worldwide. RSV is the leading cause of hospitalization for viral bronchiolitis and pneumonia in infants and children under 5 years of age [40,41]. The pathology of RSV infection includes skewing the immune response towards a Th2 phenotype and the production of IL-1β [42]. RSV induces the activation of the NLRP3 inflammasome and caspase 1, and both events are crucial for the secretion of IL-1β, IL-33, and IL-18 during infection [12,13,43,44]. Mechanistically, RSV-induced inflammasome activation requires TLR4 as signal 1 in human lung epithelial cells, while signal 2 is triggered by the RSV small hydrophobic (SH) protein, which has been classified as a viroporin and may form pores or channels on the plasma membrane [12]. During RSV infection of mouse bone marrow-derived macrophages, reactive oxygen species (ROS) and the TLR2/MyD88/NF-κB pathway mediates NLRP3/ASC inflammasome assembly, leading to the activation of caspase-1 and the release of IL-1β [13].

IL-1β was found to be increased in nasopharyngeal wash samples from infants hospitalized for RSV infection, and this was correlated with severe disease and negative outcomes [45]. Interestingly, IL-1β promotes the induction of IL-17-producing CD4 T cells [46], and higher levels of IL-17A were found in tracheal aspirate samples from infants with RSV-induced severe disease [47]. In a mouse model of RSV infection, mucus production, neutrophilia, and airway hyperresponsiveness were IL-17 dependent [48]. Collectively, these studies suggest that in both human and mice, inflammasome activation can lead to pathogenic Th17 responses during RSV infection, thereby contributing to airway immunopathology.

It has been shown that the pharmacological inhibition of NLRP3 using its selective inhibitor MCC950 or genetic deficiency of NLRP3 diminished RSV-induced immunopathology and allergic airway disease following RSV infection in mice. Targeting NLRP3 activation led to decreased IL-1β, IL-33, and CCL2 production and increased IFN-β [23]. In addition, alternative therapeutic compounds, including Chinese traditional medicine, have been reported to inhibit NLRP3 inflammasome activation, reducing RSV-induced lung inflammation [49,50]. Therefore, targeting the NLRP3 inflammasome using Chinese herbal compounds may prove valuable as a novel therapeutic approach to alleviate RSV immunopathology and rebalance the antiviral immune response in young children and babies.

## 6. Coronavirus

The viruses belonging to the genus *Betacoronavirus*—severe acute respiratory syndrome coronavirus (SARS-CoV) and SARS-CoV-2—were responsible for the 2003 outbreaks and the 2019 pandemic, respectively [51]. Different tissues such as lungs, kidneys, and liver can be compromised during coronavirus infections [52,53,54,55], which can progress to multi-organ failure. Mild to severe symptoms can be exhibited by humans but not by bats (animal host); one likely explanation is that the NLRP3 inflammasome is dampened in bats’ macrophages and dendritic cells, and therefore, they do not show signs of increased inflammation [56].

It has been proposed that inflammasome hyperactivation and the excessive production of inflammatory cytokines, known as a cytokine storm, may be linked to acute respiratory distress syndrome (ARDS) resulting from SARS-CoV-2 infection [52,57,58,59,60]. Studies suggest that the magnitude of inflammasome activation directly correlates with COVID-19 outcomes [14,60,61,62,63,64,65,66,67]. For example, elevated levels of IL-1β [14], Casp1p20 [14], IL-1RA, IL-18, and lactate dehydrogenase (LDH) [62] were detected in severe COVID-19 patients’ plasma. LDH is a recognized marker of cell death and may be released as a result of different lytic cell death modes, including necroptosis and NLRP3-mediated pyroptosis. Therefore, it is noteworthy that although elevated levels of LDH are linked to COVID-19 severity [53,54,68], LDH increase might be a result of other cell death modes associated with COVID-19. Accordingly, recent mouse studies revealed alternative pathways involved in SARS-CoV-2-induced cell death [55,59,62,63], as the treatment with the NLPR3 inhibitor MCC950 did not impact LDH levels [55,59,62]. However, the presence of inflammasome-related proteins (ASC, IL-1β, IL-18, gasdermin D) in the plasma of severe COVID-19 patients is not always correlated with unfavorable outcomes [54].

Different research groups have uncovered the molecular requirements regarding the SARS-CoV and SARS-CoV-2 structure that enable inflammasome activation. Viroporins encoded by both SARS-CoV and SARS-CoV-2 are capable of forming ion channels on host cell membranes, activating the NLRP3 inflammasome and inducing IL-1β over-expression [52,69,70,71,72]. The SARS-CoV accessory protein ORF3a promotes TNF receptor-associated factor 3 binding (TRAF3)-mediated ASC ubiquitination, leading to NLRP3 assembly [73]. Interestingly, the same viroporin has also been associated with necroptosis induction through receptor-interacting protein kinase 3 (RIPK3) [74]. Moreover, SARS-CoV ORF8b promotes NLRP3 activation through the leucine-rich repeat domain in macrophages and human epithelial cells [75]. Inflammasome and pyroptosis pathways can also be activated by other coronavirus proteins. For example, the spike protein of both SARS-CoV and SARS-CoV-2 primes the NLRP3 inflammasome in macrophages [76,77], while SARS-CoV-2 non-structural proteins (NS) are able to trigger NLRP1 and NLRP3 in different respiratory epithelial cell lines [63,78]. Thus, the recognition of a specific coronavirus protein per se promotes inflammasome activation, which can lead to hyperinflammation, as is the case of the SARS-CoV-2 nucleocapsid (N) protein. The N protein directly interacts with NLRP3, facilitating the maturation of pro-inflammatory cytokines in cultured cells and mice. More importantly, the N protein induces acute lung injury and accelerates death in mice by activating the NLRP3 inflammasome [58]. Accordingly, the inhibition of NLRP3 reduces the cytokine storm and lung injury induced by SARS-CoV-2 infection [58], suggesting that the development of NLRP3 inhibitors for clinical use in COVID-19 may prove valuable.

Nevertheless, the literature regarding the role of SARS-CoV-2 in inflammasome modulation is conflicting. While some papers report inflammasome assembly and activation, a growing body of evidence shows that SARS-CoV-2 and its isolated proteins can also dampen the expression and function of inflammasome-related proteins. For example, Ma et al. (2021) observed reduced IL-1β secretion and inhibition of pyroptosis in THP-1-derived macrophages, mouse macrophages, and primary human monocytes infected with SARS-CoV-2. The authors suggest that the N protein binds to GSDMD and blocks its cleavage, preventing GSDMD pore formation and pyroptosis [55]. In addition, Yalcinkaya et al. (2021) showed that the SARS-CoV-2 envelope (E) protein can suppress NLRP3 inflammasome priming at early stages of infection, minimizing the levels of pro-inflammatory cytokines [79]. Furthermore, SARS-CoV-2 NS-1 and NS-13 reduced caspase 1 activity and IL-1β secretion when electroporated in THP-1 cells [80]. These data could mechanistically explain the downregulation of NLRP3 and IL-1β seen in COVID-19 patients at the early stages of infection [81]. Together, these studies provide clinical and mechanistic evidence that the NLRP3 inflammasome is suppressed at the early stages of SARS-CoV-2 infection and overactivated at later stages, and this could be the main cause of the cytokine storm seen in severe COVID-19 patients. The later hyperactivation of the inflammasome could also help explain its harmful involvement in the long-COVID syndrome, as it has been reported that monocytes from patients with long-COVID syndrome display increased AIM2 inflammasome expression and responsiveness, and this was associated with lung fibrosis and worse respiratory symptoms [82].

Recent studies show that targeting inflammasome pathways may provide alternative therapeutic approaches to treat COVID-19. Sefik et al. (2022) treated SARS-CoV-2-infected MISTRG6-hACE2 humanized mice with caspase 1 and NLRP3 inhibitors and observed a better disease prognosis in the lung, with less immune cell infiltration and enhanced tissue recovery. Surprisingly, the viral load in these animals increased [61]. In contrast, treatment with MCC950 in hACE2 transgenic mice was able to reduce the viral load, as well as lung immunopathology, macrophage and neutrophil lung infiltration, and inflammatory cytokines [59]. Similar findings were obtained using NLRP3 deficient mice—reduced signs of pneumonia, lung-infiltrating inflammatory cells [59], and levels of IL-1-β and IL-6 expressed by monocytes [58]. Additionally, niclosamide treatment of SARS-CoV-2-infected human monocytes, reduced the viral load, caspase 1 expression, and IL-1β secretion. However, niclosamide treatment of SARS-CoV-2-infected K18-hACE2 mice did not affect the viral load or cytokine levels [51]. These data suggest that niclosamide has an inhibitory effect on inflammasomes and could prove valuable as a novel antiviral drug against SARS-CoV-2. In fact, prospective clinical trials using niclosamide indicated preliminary positive effects for the treatment of hospitalized COVID-19 patients [83,84].

## 7. Rhinovirus

Human rhinoviruses (HRVs) belong to the *Picornaviridae* family and *Enterovirus* genus. HRVs are positive-stranded RNA viruses responsible for 80% of common cold cases [85]. Most cases of asthma and chronic obstructive pulmonary disease (COPD) exacerbations are associated with HRV infection throughout life [85,86]. In asthmatic patients, HRV infection has been shown to increase the levels of caspase 1 [87] and upregulate the AIM2 inflammasome [88]. Moreover, the HRV 2B protein acts as a Ca^2+^ ion channel, activating the NLRP3 and NLRC5 inflammasomes [19]. Human bronchial epithelial cells highly express NLRP1, which is activated by HRV. During infection, the viral 3C protease cleaves NLRP1 between amino acids Q130 and G131, leading to inflammasome assembly, caspase 1 activation, inflammatory cell death, and cytokine secretion [89].

During HRV infection, excessive mucus production can cause airway obstruction. In human differentiated nasal epithelial cells and nasal epithelial progenitor cells, MUC5AC (airway epithelial mucin) expression is dependent on NLPR3 activation. This observation was confirmed by visualizing nasal tissue from chronic rhinosinusitis HRV-infected patients. In nasal epithelial cell models, HRV-induced inflammasome activation was dependent on the DDX33/RIG-I-NLRP3-caspase-1-GSDMD axis, inducing pyroptosis, which in turn, was essential to impair HRV replication [18]. Moreover, the inflammasome-derived cytokine IL-18 has been shown to be protective against rhinovirus-induced cold and asthma exacerbations in a human model of experimental rhinovirus infection [90]. Therefore, inflammasome-associated proteins seem to play an important antiviral role against HRV infection in humans. In contrast, IL-18 levels are not affected by HRV infection in mice [91,92]. Han et al. (2019) observed that in HRV-infected mice, macrophages are the main cells producing IL-1β and that the activation of NLRP1, NLRP3, and NLRC5 is dependent on TLR2 [92]. NLRP3- and IL-1β-deficient mice showed decreased lung inflammation and fewer infiltrating cells following HRV infection [92,93], suggesting that the inflammasome activation plays a detrimental role during HRV infection. At early stages of infection, IL-1β downregulated inflammatory cytokines and attenuated mucus production and airway hyperresponsiveness, but during later stages of infection, IL-1β had the opposite effect, which suggests that inflammasome activation at later stages of HRV infection leads to higher pathogenicity, favoring asthma exacerbations [91,92]. Furthermore, neutrophilic inflammation is involved in asthma exacerbation after HRV infection, and this effect has been reported to be driven by inflammasome activation [94].

## 8. Adenovirus

Adenoviruses are double-stranded DNA, nonenveloped viruses that usually cause mild infections in the respiratory tract, gastrointestinal tract or conjunctiva. More than 50 serotypes were described and divided into seven species of adenoviruses, from A to G. Children and adults can be infected by adenoviruses; however, the disease is more severe in immunocompromised individuals [95]. As adenoviruses induce pro-inflammatory cell death—necroptosis and to a minor extent, pyroptosis—through genetic engineering, they can be used as vectors to deliver several treatments and vaccines, which is why the interaction between the virus and the innate immune system is extensively studied [96]. Adenoviruses have been shown to activate the NLRP3 inflammasome dependent on TLR9 [97,98]. For efficient inflammasome activation by adenoviruses, cathepsin B must be released from the late endosome into the cytoplasm [97,99], with high levels of lysosomal cathepsin B correlating with superior NLRP3 activation. Additionally, the production of reactive oxygen species is required for inflammasome activation by adenoviruses [97,100].

Following repeated exposure to multiple adenovirus serotypes, a strong, long-lived humoral immunity is generated. However, adenovirus–antibody immune complexes induce the pyroptosis of human dendritic cells through the activation of the AIM2 inflammasome [21]. Moreover, antibody-opsonized adenovirus is sensed by the cytosolic antibody receptor TRIM21 (tripartite motif containing 21) and the DNA sensor cGAS/STING. Together, these sensors induce NLRP3 inflammasome formation in human monocytes without inducing cell death [101]. These studies highlight the importance of the collaboration between different sensing pathways to trigger an innate immune response against virus infection, even in the presence of a pre-existing humoral response.

Adenovirus serotype 5 (Ad5) has been shown to activate the inflammasome through the purinergic P2X7 receptor (P2X_7_R). P2X7 receptor-deficient mice showed less weight loss and decreased mortality rate and respiratory symptoms following infection compared to wild type mice. Blocking the ATP-P2X_7_R and caspase-1 pathways also protected the animals from lung immunopathology. The improved survival in P2X7 receptor-deficient mice correlated with decreased levels of IL-1β and IL-6 [102], suggesting that inflammasome activation as a result of purinergic signaling during adenovirus infection may be detrimental to the host. Darweesh et al. (2019) suggested that at later stages of the infection, Ad5 suppresses the NLRP3 inflammasome activation. The virus-associated RNAI (a small non-coding RNA) blocks NLRP3-mediated pyroptosis by inhibiting PKR and ASC interaction, preventing ASC phosphorylation and oligomerization [103].

## 9. Human Bocavirus

Human bocavirus (HBoV) belongs to the genus *Bocaparvovirus* in the family *Parvoviridae* and has different serotypes (1–4) [104,105,106]. HBoV is commonly detected in young children with acute respiratory tract illness [107]. Symptoms of the infection vary from common cold symptoms to pneumonia and bronchiolitis [108]. In adults, infection may also cause gastrointestinal symptoms. A high rate of coinfections (up to 83%) with many respiratory viruses and bacteria is commonly seen among patients infected with HBoV [109]. One of the challenges in studying HBoV is that there is no animal model of infection available; therefore, studies are largely performed using polarized human airway epithelium in an air–liquid interface in vitro [110] and consequently, the evaluation of the host immune factors involved in disease pathogenesis is limited. However, some studies have shown that HBoV infection of human airway epithelial cells promotes the thinning of the epithelium, disrupts the tight-junction barrier, and causes the loss of cilia [111,112].

HBoV1 infection induces the pyroptotic cell death of human airway epithelial cells through NLRP3 and caspase-1 activation, as knockdown of NLRP3 or caspase-1, but not caspase-3, significantly decreased cell death induced by HBoV1. HBoV1 infection also induced steady increases in IL-1α and IL-18 expression [113]. Interestingly, HBoV1 upregulated anti-apoptotic genes, that when silenced, led infected cells to undergo apoptosis. The authors hypothesized that HBoV1 switches the cell death mode from apoptosis to pyroptosis to favor a persistent infection. However, it is still unclear which viral PAMP on HBoV particles is recognized by NLRP3 to promote inflammasome activation. The lack of studies addressing the role of inflammasomes in HBoV disease pathogenesis warrants further investigation.

## 10. Non-Coding RNAs and Inflammasome Modulation during Respiratory Virus Infection

Non-coding RNAs (ncRNAs) are RNAs that are not translated into proteins or peptides [114]. Through the interaction with RNA-induced silencing complex (RISC), microRNAs (miRNAs) can prevent the translation of proteins with matching transcripts [115]. While several miRNAs have shown to inhibit NLPR3 activation [115,116], the literature regarding the expression of miRNAs during respiratory virus infections is conflicting. The overexpression of the miRNA miR-7 during influenza A virus infection of human cell lines is associated with the downregulation of inflammatory and antiviral proteins [117]. Accordingly, miR-7 has been reported to block the NLRP3 inflammasome assembly [116]. Thus, miR-7 may reduce the antiviral response against virus infections and predispose the host to develop a severe infection by blocking NLRP3. In contrast, miRNA miR-22 is downregulated in critically ill influenza A/H1N1 virus-infected patients compared to the ones with mild disease [118]. Interestingly, miR-22 has been shown to inhibit NLRP3 in the context of cancer and inflammation [115]. Therefore, miR-22 downregulation during influenza infection might lead to NLRP3 hyperactivation and increased pro-inflammatory cytokines, resulting in pronounced lung injury and disease severity. However, the role of specific miRNAs in NLRP3 regulation during respiratory virus infections has yet to be investigated.

Long non-coding RNAs (lncRNAs) regulate RNA transduction, gene expression, and chromatin remodeling and interfere in post-transcriptional events [114]. As several lncRNAs are described every day, here we will focus on the lncRNAs nuclear-enriched abundant transcript 1 (NEAT1) and metastasis-associated lung adenocarcinoma transcript-1 (MALAT1), which are associated with respiratory illness. Both NEAT1 and MALAT1 are overexpressed in severe COVID-19 [119]. Mechanistically, the lncRNA NEAT1 can be found in the cell cytoplasm colocalized with ASC, where the lncRNA stabilizes caspase 1, thus supporting inflammasome complex assembly [114,120] and promoting a pro-inflammatory environment [114,121]. The lncRNA NEAT-1 has been reported to activate NLRC4 and AIM2 inflammasomes [120]. In addition, lncRNA MALAT-1 is a competing endogenous RNA and inhibits miR-133, miR-22 [122], miR-23C, and miR-203 (micro RNAs with an NLRP3-inhibiting function) [114]. Hence, it is plausible to hypothesize that by inhibiting those miRNAs, the lncRNA MALAT-1 might indirectly promote NLRP3 activation. There are few studies investigating how ncRNAs influence inflammasome pathways during respiratory virus infections. Importantly, exploiting the association of these components may help us better understand the inflammasome regulation and its role in disease outcomes.

## 11. Conclusions

Several respiratory tract infection outbreaks occur every year. During respiratory virus infection, inflammasomes sense the acute infection and mediate a strong pro-inflammatory response to contain the infection and avoid virus spread. Inflammasome activation is also involved in lung wound healing and the restoration of homeostasis after tissue damage caused by infection. However, inflammasome hyperactivation contributes to disease pathogenesis and more severe outcomes following viral infection (Figure 1). Recent studies suggest a temporal role for inflammasomes during respiratory viral infections. While an early activation of inflammasomes is crucial to promote virus clearance, uncontrolled inflammasome activation at later stages of infection is associated with immunopathology. Therefore, understanding the interaction between respiratory viruses and the inflammasome may open new avenues for research and lead to the development of novel therapies against these infections.

**Figure 1 biology-12-00943-f001:**
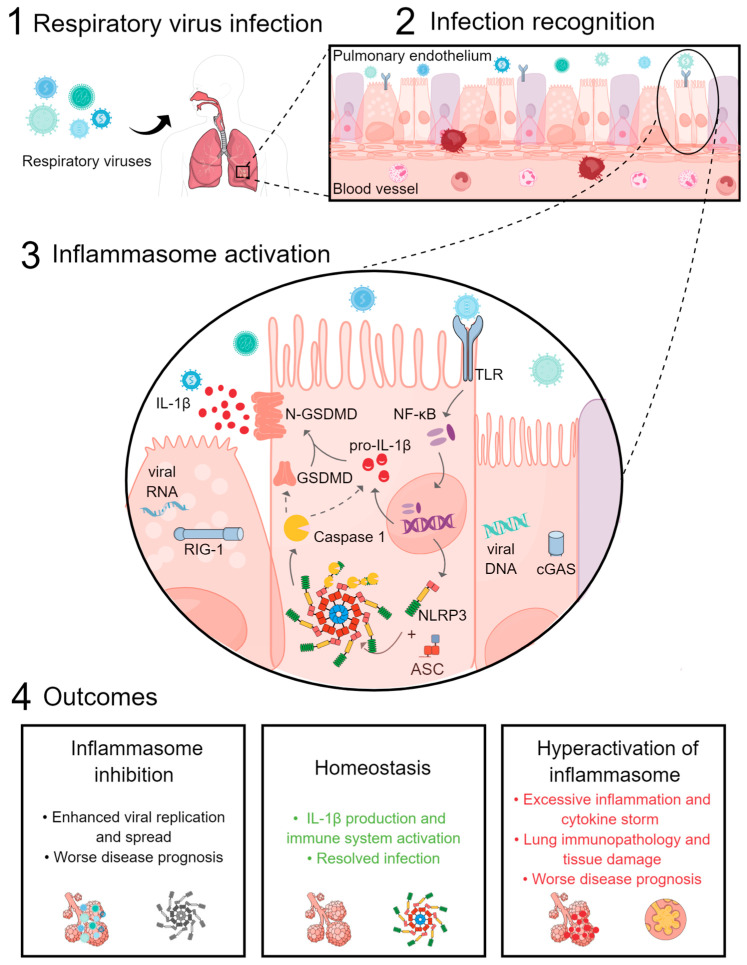
Respiratory virus activation of the inflammasome. (1) Once respiratory viruses enter the airways, (2) they encounter respiratory epithelial cells and resident immune cells, such as alveolar macrophages. Pathogen-associated molecular patterns expressed by the virus are recognized by innate sensors and activate the innate immune system. (3) Through TLR receptors, the NF-kB pathway is activated, leading to the up-regulation of NLRP3 and pro-IL-1β, which results in the inflammasome complex activation after its binding to ASC. Then, caspase-1 is cleaved and activated, which leads to the cleavage of pro-IL-1β and gasdermin D (GSDMD). Cleaved GSDMD (N-GSDMD) forms membrane pores and IL-1 β is released. The inflammasome can also be activated through cGAS and RIG-1 receptors, which recognize viral DNA and viral RNA, respectively. (4) Outcomes of inflammasome activation include inflammasome inhibition by the virus, which can easily replicate and spread, resulting in worse disease prognosis of the host. When the inflammasome is properly activated, IL-1 β production and the activation of the immune system help resolve the infection and restore lung homeostasis. Hyperactivation of the inflammasome leads to excessive inflammation and a cytokine storm, contributing to lung immunopathology, tissue damage, and worse disease prognosis. Figure created with MindtheGraph.

**Table 1 biology-12-00943-t001:** Respiratory viruses and their proteins responsible for inflammasome activation.

Virus	Activator	Inflammasome
**Adenovirus**		
Ad5	Protein VI [21]	AIM2 [21]
Ad5	dsDNA [97,98]	NLRP3 [97,98,102,103]
Ad5	dsDNA [102]	cGAS/STING-NLRP3 [102]
**Bocavirus**	Viral RNA [113]	NLRP3 [113]
**Influenza**		
IAV	dsDNA [20]	AIM2 [20]
IAV	M2 protein [27]	NLRP3 [123]
**Parainfluenza**	Viral particle [35]	TLR2/NLRP3 [35]
**Metapneumovirus**	HMPV SH [38]	NLRP3 [38,39]
**Respiratory syncytial virus**	Viroporin SH [12]	NLRP3 [12,13]
**Coronavirus**		
SARS-CoV	Spike [77]	NLRP3 [77]
SARS-CoV	ORF3a [70,73]	NLRP3 [72]/RIPK3 [74]
SARS-CoV	ORF8b [75]	NLRP3 [75]
SARS-CoV-2	Envelope [52]	NLRP3 [52]
SARS-CoV-2	ORF3a [71]	NLRP3 [71]
SARS-CoV-2	Nucleocapsid [58]	NLRP3 [58]
SARS-CoV-2	NS6 [63]	NLRP3 [63]
SARS-CoV-2	Spike [76,77]	NLRP3 [76,77]
SARS-CoV-2	NS5 [78]	NLRP1 [78]
**Rhinovirus**		
HRV	2B [19]	NLRP3/NLRC5 [19]
HRV	3C [89]	NLRP1 [89]

HMPV SH: small hydrophobic protein; NS: non-structural protein; ORF: open reading frame.

## Data Availability

Not applicable.

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
