# Peer review of "Breaking Bad: Inflammasome Activation by Respiratory Viruses"

_biology, 2023, doi:10.3390/biology12070943_

Round 1

Reviewer 1 Report

Cerato et al. provided a comprehensive summary of the impact of inflammasome activation caused by various respiratory viruses. Their well-presented article offers current literature on the interaction between respiratory viruses and the inflammasome, which could be valuable for the development of new therapeutic approaches against these infections.

Comments:

1.     The authors should clarify how their review contributes to the existing literature, considering the abundance of recent reviews on the same topic.

2.     The title of the article requires revision.

3.     Including additional information on non-coding RNAs, such as miRNAs or lncRNAs, in relation to inflammasome activation and respiratory viruses would enhance the strength of the article.

4.     Some sections of the review are lacking the author's opinion and conclusive sentence.

5.     The language in certain parts of the text needs improvement.

Language improvement required in certain sections.

Reviewer 2 Report

This review article talks about “how different respiratory viruses trigger inflammasome pathways and what harmful effects the inflammasome exerts along with its antiviral immune response during viral infection in the lungs”. It is based on sufficient research articles and carefully written. There are some minor ambiguities or suggestions I would like to discuss:

1. Line 14: the non-canonical pathways of inflammasome activation usually relates to bacterial infection. As this article mainly discusses viral infection, this part could be throwed away in case of mis-understanding.

2. Line 51: Table 1 lacks adenovirus.

3. Line 107: “inhibiting the activation”: Who, How, When

4. Line 149: This section just lists the results of several groups and is not well summarized and organized. The inflammasome activation of early/late stage should be more discussed as it links to the secret of cytokine storm and the pathological aspects of immunization.

5. Figure 1: The figure legend is well written, but the figure itself should cover more information and could be more well illustrated. For example, Figure 1(1) delivers almost nothing useful; the legend of Figure 1(2) talks about “caspase-1 is cleaved and activated, which leads to cleavage of pro-IL-1β, as well as gasdermin D”, but the figure did not show this information at all. 

Reviewer 3 Report

The review focused on Respiratory viruses and the inflammasome is limited and how the viruses trigger the inflammasome pathway. Interestingly, it described the most frequent respiratory viruses causing respiratory tract infections.

In general, the manuscript is well-written; however, I external my comments to improve the review:

In this review entitled “Respiratory virus activation of inflammasomes: the good, the bad and the ugly”, the respiratory viruses activated, or do the respiratory viruses activate the inflammasomes?

The abstract must include the research algorithm the author used for the review, e.g., PubMed, google scholar…

Influenza, Parainfluenza, Metapneumovirus, Respiratory syncytial virus, and Coronavirus infection are included in this review; nevertheless, human bocavirus is not included.

A section of perspectives must be included in this review. It must be focused on your contribution to the area of knowledge.

Check the manuscript; the authors have several errors in punctuation marks.

Round 2

Reviewer 1 Report

Appropriately revised.